# Investigating the characteristics and needs of frequently admitting hospital patients: a cross-sectional study in the UK

Reem Kayyali,[1] Gill Funnell [iD] ,[1] Bassel Odeh,[1] Anuj Sharma,[2] Yannis Katsaros,[3] Shereen Nabhani-Gebara [iD] ,[1] Barbara Pierscionek,[4] Joshua Sterling Wells,[1] John Chang[5]

¹Pharmacy, Kingston University Faculty of Science Engineering and Computing, Kingston Upon Thames, UK
²Exus Innovations, Exus, London, UK
³Medidata, London, UK
⁴School of Life Sciences and Education, Staffordshire University, Stoke-on-Trent, UK
⁵Research and Development, Croydon University Hospital, London, UK

**Correspondence to**
Professor Reem Kayyali;
r.kayyali@kingston.ac.uk

## ABSTRACT

**Objectives** This study forms the user requirements phase of the OPTIMAL project, which, through a predictive model and supportive intervention, aims to decrease early hospital readmissions. This phase aims to investigate the needs and characteristics of patients who had been admitted to hospital ≥2 times in the past 12 months.

**Setting** This was a cross-sectional study involving patients from Croydon University Hospital (CUH), London, UK.

**Participants** A total of 347 patients responded to a postal questionnaire, a response rate of 12.7%. To meet the inclusion criteria, participants needed to be aged ≥18 and have been admitted ≥2 times in the previous 12 months (August 2014–July 2015) to CUH.

**Primary and secondary outcomes** To profile patients identified as frequent admitters to assess gaps in care at discharge or post-discharge. Additionally, to understand the patients' experience of admission, discharge and post-discharge care.

**Results** The range of admissions in the past 12 months was 2–30, with a mean of 2.8. At discharge 72.4% (n=231/347) were not given a contact for out-of-hours help. Regression analysis identified patient factors that were significantly associated with frequent admissions (>2 in 12 months), which included age (p=0.008), being in receipt of care (p=0.005) and admission due to a fall (p=0.01), but not receiving polypharmacy. Post-discharge, 41.8% (n=145/347) were concerned about being readmitted to the hospital. In the first 30 days after discharge, over half of patients (54.5% n=189/347) had no contact from a healthcare professional.

**Conclusion** Considering that social care needs were more of a determinant of admission risk than medical needs, rectifying the lack of integration, communication and the under-utilisation of existing patient services could prevent avoidable problems during the transition of care and help decrease the likelihood of hospital readmission.

## INTRODUCTION

A desire to reduce the increasing cost of healthcare provision is an impetus for many countries to search for new ways to both increase efficiency and improve the quality

### Strengths and limitations of this study

► The study evaluated the patients' journey from admission, discharge to post-discharge, providing a holistic picture of patients' experiences.
► The study successfully implemented a cross-sectional questionnaire across a diverse sample population using a postal survey method with no reminders sent.
► The target sample included all patients ≥18 years of age who experienced ≥2 admissions in the past 12 months at Croydon University Hospital (CUH).
► The study used linear regression analysis to identify significant contributing factors to patients being admitted >2 times in a 12-month period.
► The study is representative of patients admitted only to CUH and is limited by the memory of the respondents.

of hospital care. Reducing the cost of early hospital readmissions is an objective with clear benefits for both providers and patients.[1]

In the UK, readmissions were estimated to cost the National Health Service £2.4 billion in 2012–2013, which is 19% of the total emergency admission cost of £12.5 billion.[2] Since 2011, UK hospitals have been financially penalised for patient readmissions occurring within 30 days of discharge, which is considered as early readmission.[3]

The UK financial penalty was introduced in 2011 to discourage hospitals from attempting to free up beds by discharging patients before they were ready.[3] However, not all early readmissions are due to suboptimal patient care and many readmissions may be unavoidable and appropriate, for example where patients are chronically or terminally ill.[4 5] Two UK studies found around 60% of early readmissions were due to the same reason as the primary admission, suggesting that

these could have been reduced by medication reviews, better discharge communication and a rapid response to preventable issues.[6 7]

Both polypharmacy and chronic conditions such as chronic obstructive pulmonary disease (COPD), cardiovascular disease and diabetes have been found to be associated with readmission rates and increased needs following discharge.[8 9] Accurately identifying patients as high risk enables resources to be channelled specifically to these patients through supportive interventions, rather than providing for all patients, many of whom may not be at risk of readmission. Several predictive models have been developed in the UK such as Patients at Risk of Readmission within 30 days [10] and in Canada the [11]Length of Stay, Acuity, Comorbidities, ER with relatively good predictive accuracy.

Evaluating the effectiveness of interventions designed to prevent early readmissions is problematic due to the lack of robust studies with good methodologies.[9] Intervention types which have been studied, often in combination include: extensive discharge planning, telephone calls, home visits, a 24-hour hot line and patient education.[9] The provision of follow-up telephone calls is a common intervention, with variation in the number and length of calls and profession of caller. The most successful results included both pre-discharge and post-discharge interventions.[12]

Schemes for supporting patients with their medications in the community were introduced into community pharmacies in 2005. Medicine Use Reviews (MURs) and New Medicines Service (NMS) can support patients with medication adherence as well as identifying interactions and other problems. The NMS is specifically targeted at patients with long-term conditions such as COPD to support patients starting a new medication.[13]

The need for successful management of the pre-discharge and post-discharge period is highlighted in the National Institute for Health and Care Excellence Guidelines,[14] developed in 2015 to help with the transition of adult patients with social care needs from hospital to the community. These guidelines emphasise the importance of the transition of care being coordinated using good communication. All healthcare professionals (HCPs) involved with the care of the patient in hospital and the community, should be included in the communication loop, with all patients/carers being provided with a medication list and a care plan with a single HCP responsible for coordinating the discharge for both social and medical needs.

This paper reports on the first stage of the OPTIMAL project,[15] funded by Innovate UK. The OPTIMAL project encompasses the development of a predictive risk model, together with a supportive post-discharge patient intervention with the aim of reducing early hospital readmission. Although the success of both predictive risk models and interventions to prevent hospital readmission have been developed and studied separately before, this is the first time, to our knowledge, that a predictive model and a preventative intervention have been integrated to support patients.

The aim of this study was to undertake a needs assessment to investigate any common characteristics of patients admitted more than one time to CUH in a period of 12 months and understand their experiences of both the discharge process and the immediate post-discharge period. The study also sought to determine factors contributing to frequent admission (>2 in 12 months). This will assist in the development of an appropriate post-discharge intervention for patients identified at high risk of readmission.

## METHODS

A cross-sectional study was carried out at CUH. Patients were considered for inclusion in the study if they met the following criteria: ≥18 years, a home address on the CUH database, experienced ≥2 admissions to CUH in the past 12 months (August 2014–July 2015). Paediatric, oncology and maternity patients were excluded from the study. CUH research and development (R&D) department using patient records identified a total of 2722 patients who met the inclusion criteria. To provide a confidence level of 95% and a CI of 5%, the sample size was calculated as 337 patients. As a low response rate may be expected from postal survey, all 2722 patients were invited to complete the postal questionnaire (online supplementary file). An explanatory letter was sent with the questionnaire together with a prepaid return envelope. The questionnaire was only made available in English and no reminders were sent.

A quantitative cross-sectional questionnaire survey was designed using a mixture of open and closed questions. The validated tools AUDIT-C (a brief alcohol screening tool used to identify alcohol dependency)[16] and a medical health literacy score[17] were incorporated together with other questions which investigated patient experience and knowledge of medication and discharge counselling. The questionnaire was in four sections: first, demographic information, collecting personal information such as age, as well as medication list and current medical conditions. Second, understanding the patient's admission experience, the reason for the patient's attendance at accident and emergency (A&E) and satisfaction with the admission process. Third, the patient's discharge experience, investigating patients' involvement in their discharge planning and the provision of medication counselling. Finally, understanding the patients' post-discharge experience, the discharge support received by patients, as well as patients' confidence in managing their health and coping at home post-discharge. The experience sought was based on the patient's most recent admission.

### Pilot

After receiving ethics approval, a pilot study was conducted which involved asking 10 patients from the discharge lounge at CUH to complete the survey for

validation. Minor changes were made to the question-naire. To prevent any bias, the findings from the pilot were not included in the final results.

## Patient and public involvement

The study was a follow-up study from 50 patients at the Trust who indicated mixed experience in counselling and shared decision-making during admission. As part of the funding, the researchers agreed to inform patients/public of the outcome of the study. This was completed via the public engagement forums within the Trust.

## Data analysis

The responses from the returned questionnaires were analysed using IBM SPSS V.23 through descriptive statistics and the $\chi^2$ test for independence, with a level of significance set at 5% (p<0.05). A comorbidity poly-pharmacy score (CPS) was calculated (defined as the total of the number of pretrauma comorbidities and the number of preadmission medications in trauma patients ≥45 years). Our modified calculation was performed for all patients ≥45 years, using the number of medications specified in the questionnaire, together with the number of existing comorbidities recorded. A 3-question AUDIT-C score[16] was calculated, with each question having a possible score of 0–4 and giving a total score in the range between 0 and 12. A score of ≥5 is considered positive, indicating a higher risk of alcohol consumption. A single question health literacy tool was used giving scores of 1–5, with scores >2 indicating some difficulty reading printed health material.[17] The number of medications most associated with adverse drug reactions (ADR) resulting in hospital admission was also recorded for each patient.

A linear regression analysis was carried out on the data to help identify significant patient characteristics which may have contributed to a greater number of admissions in the previous 12 months. This was carried out by adding a dependent variable column 'frequent_admitter' to the data which was then assigned 1 if a patient's admissions in the previous year were >2 or 0 if ≤2. The independent variables included in the regression analysis were: admission reason, ethnicity, condition complexity indicator (which was set if a patient described their existing situation as complex/complicated or reported ≥2 conditions), a care indicator (identified by patients who were in receipt of some home care), CPS, patient age, gender and number of medications. The linear regression was repeated to understand the contributors to very frequent admissions, which was >3 in the previous 12 months and for those over 55 years of age with >2 admissions in the previous 12 months. Any row where any of these variables were missing was excluded, thus leaving 169 patients to be included in the regression analysis. This number was 137 patients when only including those greater than 55 years of age in the regression analysis.

## RESULTS

The questionnaires were sent to 2722 patients, 347 were completed and returned giving a response rate of 12.7%.

The most common reasons given for the last admission were respiratory problems such as asthma and COPD (15.0%, n=52). Nearly 10% (n=33) of patients were admitted due to a fall. Nearly a third (n=101) of patients reported more than one condition or described their condition as complex (table 1).

Over a quarter (28.8%, n=99/344) of patients lived alone and less than 5% (4.4% n=15/344) lived in a care home. Not all patients had someone to care for them; 26.7% (n=88/330) reported that they had no available care. Only 13.1% (n=43/328) of patients currently smoked, which is less than the UK average of 19%.[18] However, 39.3% (n=129/328) described themselves as ex-smokers. Nearly a third of patients had a limited health literacy score (29.8%, n=101/339) and over 15% (16.6%, 30/180) had a positive AUDIT-C score associated with a higher alcohol consumption risk.

### Admission

Over half of patients were referred to A&E by an HCP (58.8%, n=204/347), with just over a third (34.6%, n=120/347) of patients reporting that a family member or they themselves made the decision. Although, two-thirds of patients (67.4%, n=234/347) were consulted regarding admission and care decisions, most patients (89.6%, n=311/347) wanted to be more involved with these decisions. The most frequently expressed comments about the admission experience concerned communication problems and the lack of provision of information (41.1%, n=35/85).

### Regression analysis

Five variables were found to be significantly associated with >2 admissions in the previous 12 months. These were admission for a fall (p=0.01), not identifying as having a complex condition or reporting <2 conditions (p=0.003), age (p=0.008), male gender (p=0.007) and being in receipt of care at home (p=0.005). Additionally, the overall regression is significant according to the F test (F=0.04). These factors were still significant for the sample when analysing only those patients ≥55 years of age (F=0.007). The only change was that admission due to infection became significant in this sample (p=0.002). For patients with admissions >3 in 12 months CPS was found to be an additionally significant factor (p=0.02). All other independent variables were not found to have a statistically significant contribution to the frequency of admission.

### Discharge

Nearly half of patients, (42.1%, n=146/347) were not informed of the discharge decision 24 hours in advance, including 43.4% (n=43/99) of those who lived alone.

Over half of patients (54.0%, n=187/347) were discharged from the hospital on a weekday between 12:00

**Table 1** Demographics and medical conditions of respondents

| Parameter | n (%) | Mean (SD) | Range | Mode |
|---|---|---|---|---|
| Age (n=334) | 334 (100.0) | 69.2 (18.2) | 18–100 | 84 |
| Gender (n=337) | | | | |
| Male | 155 (46.0) | | | |
| Female | 182 (54.0) | | | |
| Ethnicity (n=333) | | | | |
| White | 250 (75.1) | | | |
| Black | 34 (10.2) | | | |
| Asian | 25 (7.5) | | | |
| Other | 24 (7.2) | | | |
| Medical history | | | | |
| No. of admission in previous 12 months* | 347 (100.0) | 2.8 (1.9) | 2–30 | 2 |
| No. of admission in previous 30 days* | 32 (10.8) | 1.4 (0.9) | 0–6 | 1 |
| Most common reason for last admission (n=347) | | | | |
| Respiratory conditions | 52 (15.0) | | | |
| Chest pain | 18 (5.2) | | | |
| Other pain | 20 (5.8) | | | |
| Fall | 33 (9.5) | | | |
| Infections excl. chest | 28 (8.1) | | | |
| Cardiac conditions | 23 (6.2) | | | |
| Other | 132 (38.0) | | | |
| Not specified | 41 (11.8) | | | |
| Most common existing medical conditions (n=347) | | | | |
| Cardiac conditions | 59 (17.0) | | | |
| Respiratory conditions | 52 (15.0) | | | |
| Hypertension | 41 (11.8) | | | |
| Diabetes | 42 (12.1) | | | |
| None specified | 123 (35.4) | | | |
| >1 Long-term condition or described as complex | 101 (29.1) | | | |

*Number of patients admitted within previous 12 months and 30 days. Mean (SD), range and mode reflect number of admissions per patient sample.

and 18:00. However, about a quarter of patients (21.3%, n=74/347) were discharged between 18:00 and 06:00 with 17.6% (n=13/74) of them living alone with an average age of 71.2 years.

Two-thirds (67.4% n=234/347) of patients agreed that the decisions regarding the discharge procedure were clearly explained (table 2). However, only a third of patients (34.3%, n=119/347) were provided with information to enable them to detect signs of deteriorating health. Furthermore, only a third of patients (33.4% n=116/347) were provided with contacts for out-of-hours support. Less than a third of patients were referred to a post-discharge service and less than half of respondents reported joining this service (table 2).

When patients were asked their opinion about their discharge procedure, 72 patients responded. The main concerns expressed were the poor provision of information and communication difficulties at all levels. Patients' concerns included the lack of communication between hospital staff and the patients/patients' families (48.6%, n=35/72), including two elderly patients discharged without informing their families. One patient stated, 'More co-ordination is needed between the pharmacy and wards.' Patients were also concerned about long waiting times (36.1%, n=26/72), with 42.3% (n=11/26) of the waiting times involving a delay in receiving medications.

## Medications
Two-thirds of patients reported taking at least one regular medication (67.4%, n=234/347). Three-quarters of these patients experienced changes to their medications while in hospital (75.2%, n=176/234), but over a quarter of these patients (28.4%, n=50/176) did not

| Table 2 | Patients' discharge experience |
| --- | --- |
| **Patient discharge experience (n=347)** | **n (%)** |
| Received discharge information from a doctor | 188, (54.2) |
| Felt the decisions at discharge were clearly explained | 234, (67.4) |
| Was fully consulted in the decision of being discharged | 226, (65.1) |
| Received a written copy of care plan | 146, (42.1) |
| Told about signs or signals to watch out for indicating health was worsening | 119, (34.3) |
| Told who to contact if health deteriorated | 84, (24.2) |
| Told who to contact for out-of-hours help | 116, (33.4) |
| Referred to a post-discharge service | 95, (27.4) |
| Patient joined the post-discharge service (n=95) | 46, (48.4) |
| Provided with details of local support groups | 63, (18.2) |

| Table 4 | High-risk drugs |
| --- | --- |
| **Number of high-risk medicines (n=234)** | **n (%)** |
| >5 | 2, (0.9) |
| 5 | 15, (6.4) |
| 4 | 14, (5.9) |
| 3 | 49, (20.9) |
| 2 | 52, (22.2) |
| 1 | 52, (22.2) |
| 0 | 50, (21.4) |

receive any counselling. Over two-thirds of patients (70.5%, n=165/234) agreed that medication information was explained in a way they could understand. However, 34.6% (n=81/234) would have liked more information regarding their medications.

The average number of medications per patient was 4.2 with of a range of 0–25. Nearly two-thirds (65.0%, n=152/234) of patients were taking ≥5 medications. The most commonly prescribed medication classes are shown in table 3.

Some of the medication combinations found are not routinely recommended, due to being identified as risky.[19] For example, 10.7% (n=25/234) of patients were taking the high-risk combination of two or more antiplatelet drugs or an antiplatelet drug together with the anticoagulant warfarin. Also 4.3% (n=10/234) were taking the high-risk triple combination of (ACE inhibitors (ACEI)/angiotensin receptor blockers (ARB)), a non-steroidal anti-inflammatory drug (NSAID) and a diuretic.

| Table 3 | Most common medication classes |
| --- | --- |
| **Medication class (n=234)** | **n (%)** |
| Proton pump inhibitors | 107, (45.3) |
| Statins | 105, (44.5) |
| Antiplatelet drug | 84, (35.6) |
| ACE inhibitors/angiotensin receptor blockers | 80, (33.9) |
| Beta blockers | 80, (33.9) |
| Calcium channel blockers | 65, (27.5) |
| Loop diuretics | 52, (22.0) |
| Opioid analgesics (including tramadol) | 32, (13.6) |
| Oral anticoagulants | 34, (14.4) |
| B-2 agonists | 35, (14.8) |

Over half of patients (56.4%, n=132/234) were prescribed two or more than two medications that could put them at high risk of admission due to an ADR (table 4).[20]

The average calculated CPS score was 7.5. Scores greater than 7 are associated with an increased risk of falls and length of hospital stay, complications, short term and 1 year mortality,[21] over 40% of patients (n=135/313) had scores >7 and 20 patients were considered as severe or morbid with scores between 15 and 32.

### Post-discharge experience

While 70.3% (n=244/347) of patients were confident in managing their own health, 41.8% (n=145/347) had concerns about being readmitted to the hospital, with two patients feeling that their last admission was due to medicine errors that could have been avoided. Receiving medication counselling in hospital (55.1%, n=191/347) was significantly associated with patients feeling more confident in the management of their healthcare issues (p=0.013). Three-quarters of patients (74.9%, n=260/347) were confident in managing their supply of medicines, but were less confident in managing their social care issues (34.3%, n=119/347) and healthcare issues (48.9%, n=170/347).

Almost half of patients (46.9%, n=163/347) were very satisfied or satisfied with the available support post-discharge in managing their health needs. However, less than a third of patients were satisfied (27.4%, n=95/347) with the support for their social care needs.

During the crucial first 30 days post-discharge from hospital, over half (54.5%, n=189/347) of patients did not receive any contact from a hospital, general practitioner (GP), pharmacy or other post-discharge services. Only 17.6% (n=61/347) of patients reported being contacted by their GP. During this time, patients were also very reticent to contact an HCP themselves, with only 12.1% (n=42/347) of patients reporting initiating contact.

Just under a quarter of patients (24.2%, n=84/347) were contacted by other post-discharge services, of the 58 patients who specified a service, half were contacted by community or other nurse, but only 15% (n=3/20) of patients suffering from COPD, 13 of which were admitted with a respiratory problem/exacerbation, were referred

to the respiratory HOT clinic (a rapid access clinic to help patients with COPD avoid hospitalisation).[22]

Community pharmacy support services were not well used post-discharge and only 4.0% (n=17/347) of patients were referred to MUR, with 69.7% (n=242/347) of patients being unaware of MUR services. However, 50.4% (n=175/347) of patients were interested in receiving this service. Similarly, 78.9% (n=274/347) of patients were not referred to NMS, with 51.6% (n=179/347) of patients being interested in receiving this service.

## DISCUSSION

This questionnaire-based study followed patients that had ≥2 hospital admissions/year living in the vicinity of CUH from admission, through discharge to post-discharge. Despite the low response rate, this is the first study that captures the complete patient journey from admission, discharge, through to post-discharge care. Furthermore, it identified characteristics of patients with high admission rates. A strength of this is the holistic nature of the reported data, which provide a comprehensive picture of these patients' experience of the support they were given, their physical health and medication when discharged from hospital. The data highlight a wide range of areas for improving patient support, including communication, utilisation and integration of services and medication counselling.

The study had several limitations: First, it is representative of the population around CUH and admissions to that Trust only, as well as being limited by the memory of the respondents. Second, not all patients fully completed the questionnaire, hence, statistical significance was not achieved for the whole questionnaire. Third, as the questionnaire was only available in English, this limited the study to participants who had sufficient English, the black population was also under-represented at 10.2% compared with the 2011 census figure of 20.2%.[23] Three quarters (75.1%) of patients described themselves ethnically as white, which is an over-representation when compared with the Croydon borough 2011 census figure of 47.3%.[23]

Regression analysis identified five patient characteristics associated with higher admission. It is interesting that two of these factors: falls and being in receipt of care, both require liaison with other services post-discharge to provide adequate support in the patient's home. Suffering from falls is a well-known cause of hospital admission and corroborates with other studies,[24 25] but being in receipt of care is, as far as we are aware a novel, though not surprising reason for admission. The male gender has previously been associated with increased admission, specifically in older people, which is pertinent to our study given the mean age of 69.2 years among participants.[26] Falling was the second most common reason for admission as reported by nearly 10% of patients. Polypharmacy, higher CPS score and identifying one's condition as complex or having >2 existing conditions were not

significantly associated with >2 admissions in 12 months. However, a higher CPS score was found to be a significant contributor to high levels of admission (>3 in 12 months). Medications may often be implicated in falls with an increased risk for patients even those taking <5 medications, however the medication class may be deemed to be more significant than the number.[27 28] Nevertheless, a higher CPS has been associated with an increase in falls by other studies, which may explain why this study found this factor to be significant for those that had >3 admissions in 12 months.[29 30] Nearly 50% of patients had a CPS score ≥7% and 65% were taking five or more medications. An Australian study observed a median increase from 3 to 6 annual attendances in the emergency department (ED) for those ≥65 years old who presented with comorbidities and polypharmacy (≥5 medications), among other factors.[29]

There is additional evidence to suggest that comorbidities are a significant factor when predicting early readmission. The Charlson Index, which predicts 10-year mortality based on patients' comorbidities, was found to be significantly associated with readmission within 28 days for patients scoring ≥3 in a retrospective observational study by Li et al.[30] Interestingly, Considine et al[31] found that comorbidities were not significant predictors of readmission ≤1 day post-discharge for patients from acute-care, however health service use was notable in the 6 months preceding the index admission with ≥1 ED attendance or ≥1 hospital admission in 42.6% (n=579) and 40.7% (n=553), respectively. Although our study focused primarily on frequent admission as opposed to readmission, the latter study could provide an explanation of why comorbidities were only a predictor of high admission rate (>3 in 12 months).[31]

It must be noted that in this study, medications and conditions were self-reported. However, these were not found to be significantly associated with frequent admission (>2 in 12 months), thus highlighting that social care needs are superseding medical needs in determining increased admission risk with medical needs becoming significant in those with >3 admissions in 12 months.

Receipt of medication counselling was significantly associated with patient confidence in managing health (p=0.013). Medicine combinations were reported which could have been questioned, such as patients taking two antiplatelet drugs or an antiplatelet drug with warfarin, which can lead to an increased risk of bleeding.[19] Ten patients were taking the combination of NSAID, ACEI/ARB together with a diuretic, this combination is associated with an increased risk of acute kidney injury.[32] Community pharmacists being the most accessible HCP, are well placed to identify medications which cause adverse events to patients and increase their risk of falls. Patients were not referred to and had a lack of awareness of community pharmacy medicine information schemes—MUR or NMS. This was a missed opportunity for medication support post-discharge in the community. In fact, an initiative at CUH that piloted the provision of

domiciliary MUR to housebound 'high-risk' patients by community pharmacists resulted in reported avoidance of hospitalisation.[33 34]

Although nearly three quarters of patients felt consulted in the decisions leading to their discharge, patients expressed dissatisfaction with the discharge process, with long waiting times, delays and poor communication reported as the most common complaints. These findings correlate with an AGE UK report[35] investigating older people's experience of hospital readmission. Delays in discharge and lack of information are upsetting and confusing. Patients should at least be provided with updates as to the progress of their discharge. Although this study is limited to the experiences of the population around Croydon, a study from Liverpool Hospital UK[36] reported similar percentages of patients (70%) who felt that discharge decisions were explained, with the long wait for discharge medications also having a negative influence on the discharge experience.

Nearly 50% of patients were worried about being readmitted to hospital and commented on finding the experience stressful and wanting to avoid readmission. Good communication and information sharing supports the transition from hospital and helps prevent readmission.[14 37] Contact information should be provided in case of a short-term crisis, which should be proactive rather than waiting for a more serious problem to arise. However, it was found that nearly 40% of patients were not provided with the signs of deterioration of their condition and nearly three-quarters of patients were not provided with details of who to contact if this situation arose. This lack of information could result in patients returning to hospital. Additionally, patients' carers and families were not always informed of the discharge, making it hard for them to adequately support the patient at home.

Poor integration of services was found both within the hospital and between primary and secondary care providers. Patients with social care needs should be contacted by a GP or community nurse within 24–72 hours of discharge.[14] However, less than 20% of patients were contacted by their GP within 30 days of discharge. A further 12.1% contacted an HCP themselves. Additionally, patients were not being referred to post-discharge services which could have supported them. Despite 20 patients reporting suffering from COPD and 13 of these patients reporting respiratory problems/exacerbation as the reason for admission, only 3 patients were referred to the respiratory HOT clinic at CUH[22] which provides an integrated team of multidisciplinary HCPs. Nearly one-third of patients were dissatisfied with their social care, thus it is not surprising that those receiving care were more at risk of frequent admission. A lack of transition of care was reported, with a need for low level practical support during the first few days after discharge. This is a shared outcome with the AGE UK report.[35]

More integrated support such as that provided by Lewisham Integrated Medicines Optimisation Scheme[38] can break through traditional boundaries of care, but as these authors note such links with services take time to build. With an increasing ageing population with more multi-morbidities, the integration of service delivery across different clinical areas becomes more important to provide appropriate individual care, rather than the current disease-focused practice.[39] A move to a shared responsibility, is required across multiple areas—social, voluntary and clinical—to provide the integrated personalised care that patients need.[40]

## CONCLUSIONS

The study highlighted gaps in care during the patient discharge journey. Admission for a fall and receipt of care were significantly associated with higher admission rates. Additionally, it reports for the first time, that social care is an important determinant of frequent admission (>2 in 12 months) in a predominantly older population. Before discharge, patients lacked medication counselling, information on symptoms of deteriorating health or HCP to contact if this situation arose. An improvement in communications at all levels would benefit patients, ensuring patients are informed of delays and decisions. Additionally, patients' confidence in their care being well managed may be increased by demonstrating that communication channels are open between different HCPs. Post-discharge, patients were lacking referrals to relevant services which could have supported them. The study highlighted that transitional care is fragmented between different services of primary, secondary and social care as well as the voluntary sector. This lack of integration is causing patients avoidable difficulties. Improvement could be made by increasing HCP awareness of the available services, both voluntary and statutory, in the local area and encouraging links. Integrating services would increase the utilisation of existing resources, such as community pharmacy medicine support schemes, hospital services, for example, respiratory HOT clinics as well as voluntary services, with care pathways using all relevant services across each sector.

**Contributors** RK was the principal investigator of the study. She was responsible for the design of the study. She also organised and coordinated all aspects of this research. GF worked alongside RK to draft the publication. BO contributed to data collection. The analysis of the results was carried out by RK, GF, BP, SN-G, YK, AS, JSW and JC.

**Funding** This research was carried out as part of the OPTIMAL project which has received funding from Innovate UK.

**Competing interests** None declared.

**Patient consent for publication** Not required.

**Ethics approval** From Kingston University Delegated Research Ethics Committee (Ref: 1415/035) and approved by the R&D department by CUH as a service evaluation.

**Provenance and peer review** Not commissioned; externally peer reviewed.

**Data availability statement** No data are available.

**ORCID iDs**
Gill Funnell http://orcid.org/0000-0001-8816-7523
Shereen Nabhani-Gebara http://orcid.org/0000-0002-5784-4779

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
