## [Reviewer comments · BMJ Open]

ARTICLE DETAILS

TITLE (PROVISIONAL)	Investigating the Characteristics and Needs of Frequently Admitting Hospital Patients- A Cross- Sectional Study in the United Kingdom
AUTHORS	Kayyali, Reem; Funnell, Gill; Odeh, Bassel; Sharma, Anuj; Katsaros, Yannis; Nabhani-Gebara, Shereen; Pierscionek, Barbara; Wells, Joshua; Chang, John

VERSION 1 – REVIEW

REVIEWER	Sushant Joshi University of Southern California, USA
REVIEW RETURNED	24-Jan-2020

GENERAL COMMENTS	Thank you for the opportunity to review this manuscript. This paper contributes to the literature on the reasons for early hospital readmissions and what can be done to reduce it. This paper tries to understand the experience of admission, discharge, and post-discharge care. There is a focus on reducing readmissions. Hospital readmissions are costly and burdensome to patients. Hospital readmissions can suggest a lack of quality of care during initial admission as well as a lack of integration of other health care services. Given this, there has been a number of studies that have tried to understand the reasons for early readmissions as well as avenues to reduce readmissions. Their findings suggest that patients with social care needs were more likely to be readmitted than those with complex medical needs. With that said I have some comments below: 1. Studies usually distinguish between early readmissions and late readmissions after discharge. It is generally assumed that early readmissions, like 30-day post-discharge, is a better signal of quality of care received for the index hospitalization.2. The study does not make the distinction between planned and unplanned readmission. Is it possible to distinguish this and include cases of only unplanned readmission rather than combine both planned and unplanned readmissions?3. Does the study only consider individuals who were admitted to the same hospital? Do we know anything about individuals who might be readmitted to other hospitals within the same area and if this affects the results of the analysis?4. Page 7, line 19 delete the researchers5. Page 8, can length of stay during index hospitalization be included as an independent variable in the regression analysis? Length of stay is generally used to predict readmissions.6. Similarly, can regression analysis be run for elder patients (for e.g. 55 and above) only? Given the mean age of 69.2, individuals with low age might be outliers in the data.7. Page 8, Line 59, /333 is missing for the row Prefer not to say.
--

	8. Page 10. On the regression results are the coefficients meaningful?
--	--

REVIEWER	Maryann Street Deakin University and Eastern Health, Australia
REVIEW RETURNED	14-Feb-2020

GENERAL COMMENTS	Thank you for the opportunity to review this clearly written and well presented manuscript. It adds another dimension to the body of research in the area of frequent re-admissions to acute care. I hope you will find the comments and suggestions helpful (see below): TITLE and ABSTRACT There is some confusion for the reader in the term 'frequently readmitting' in the title and the inclusion criteria for the study cohort. Please confirm whether it is ≥ 2 admissions or ≥ 2 readmissions in 12 months. The aims suggest that participants had at least 3 admissions, but the inclusion criteria in participants section clearly state ≥ 2 admissions. The results also show the range for number of admissions started at 2. This is an important point for clarification because 2 admissions in 12 months would not normally be regarded as frequent. P2 lines 45- 52 – See comments below about regression analysis in the Results section. INTRODUCTION P5, line 29 - This study is described as a phase of the OPTIMAL Project, funded by Innovate UK. A reference should be provided, either to the protocol for the OPTIMAL project, or a weblink to the funding body, where project is referenced. P5 Line 45 – frequent admitters (≥ 2 in 12 months), not readmissions? METHODS P6, lines50-59 – Surveys included questions about the patient's discharge experience and post-discharge experience. It is not clear which admission / readmission the survey participant were asked about? P7, lines19 and 21, the term 'the researchers' is repeated. P7 line 42 – suggest replacing the word 'complaints' with comorbidities, as that is what has been described in the preceding section. P7 line56 to P8 line 19 – If the regression was conducted as described this is not linear regression, but logistic regression. Logistic regression is used to model a binary dependent variable (in this case 'frequent-admitter' –Yes or No). The definition of 'frequent admitter' is described as a patient who had 3 or more admissions in 12 months, compared to a non-frequent admitter who had 2 admissions in 12 months. This comparison is possibly too small a difference for logistic regression, especially as the final number of patients included in the regression analysis was 169, for 7 independent variables. Further comments about the regression analysis are provided in results section. Perhaps this data was not recorded, but were the readmissions for the same diagnosis / condition as the index admission? What was the median time between admissions? What proportion of readmissions were within 30 days? RESULTS The format of the tables could be improved. Generally horizontal lines are not required throughout, just at the top and bottom of the header and table. The columns are for the parameter / variable
--

and then number and percentage (n, %) or mean and standard deviation (mean, SD) or median and interquartile range (mdn, IQR). It was noted that the denominator changed frequently for different items, and when this occurs, the number relevant for that item can be written after the variable name.

In Table 1, Ethnicity, the number and percentage for 'Prefer not to say' was not in the same format as items above.

Throughout the results, the denominator used to calculate percentages changed frequently. I assume this was due to missing data when respondents did not answer some questions. This means that some statements misrepresent the true percentage of all respondents. For example, the total number of surveys returned was 347, the number reporting age is 334, gender is 337, ethnicity 333 etc..... In most instances, the authors have made this clear.

However, in the section on Discharge, the statements are confusing. It would be much clearer if the same denominator was used throughout this section – either 347 as total number of respondents, or 334 as the number who completed the questions relating to discharge. In particular, the statement on line 46 – 'nearly three quarters of patients (72.4%, n=231) were not provided with contacts for out-of-hours support' is misleading as $231/347 = 66.6\%$. This misrepresentation of percentages is also evident in Table 2; the number of respondents who were told who to contact if their health deteriorated was 84, that is 24.2% of all respondents, but is reported as 70.6% (84/119). It is not clear why 119 was used as the denominator for some items in Table 2, but this over-represents the proportion of respondents for those items. The item below this in the table is written in the negative – 'Not told who to contact for our-of-hours help', which is inconsistent with the item above. Please review and revise this table for consistency.

Similarly, in the section on Medications, the number of patients (N) is recorded as 236. But this does not match the text, where the first sentence has N=341, and even removing the 13 patients who reported taking no medications, does not result in N=236? Perhaps the denominator for Table 3 could be the total number of medications reported.

The median time to readmission from index admission and the discharge destination were not reported, but are of interest to the reader.

P10 lines 14-21 Regression analysis- there is insufficient detail provided to assess the validity of the results from the regression analysis. No univariate analysis has been provided for the items included in the regression. If the regression was conducted as described in the methods, the items which were found to be significant were for increased risk of hospital admission (compared to those who had 2 admissions in 12 months). As such, the variables of admission related to a fall, being in receipt of care at home are logical. However, the item for not having a complex condition or reporting less than 2 conditions being associated with increased risk of hospital admission is not logical. For the regression analysis to be included, further details need to be provided.

P13, line8-12 – Post-discharge experience - It was surprising that in a series of statements of results from the survey, one p-value is given. This suggests that Chi-square analysis was done comparing groups of respondents who were / were not confident in managing their own health. It is difficult to assess the validity of this statistical result with the limited information provided.

	P13, line 45; suggest this statement about referral to HOT clinic be prefaced by stating the number of patients this would have been an appropriate, relevant referral, as described in the discussion. DISCUSSION While the discussion is clearly written, the description of the study findings in relation to current literature was limited. P14, lines 14-16: To the authors knowledge this may be the first study to capture the complete patient journey across all ages. However, there is an Australian study published in 2019 by Berry et al., that looked at this issue for older patients. See full reference below. P15, lines 17-32 – ‘in this study medications and conditions (comorbidities) ...did not act as predictors of readmission.’ This would seem to be contrary to other published studies, such as that by Considine et al., 2019 that found the presence of comorbidities was a significant feature for adult medical patients and when controlled for confounders, the number of comorbidities increase the risk of unplanned readmissions within 28 days. Further a study by Li et al, 2015 on the factors associated with unplanned readmission found a Charlson index ≥ 3 was independently associated with increased risk of 28 day admission. (references below) REFERENCES A very consistent style of referencing has been used, however the in-text citations do not always conform with the guidelines for authors. For example p14 lines 41 and 45, the in-text reference [22] is not immediately following the punctuation but has a space, and then has no space before the start of the following sentence. Berry D, Street M, Considine J. (in press) Service use by older very frequent emergency department users: A retrospective cohort study. Australasian Emergency Care. Accepted 18th May 2019. Considine J, Fox K, Plunkett D, O'Reilly M, Darzins P. (2019). Factors associated with unplanned readmissions in a major Australian health service. Australian Health Review, 43(1), 1-9. doi:http://dx.doi.org.ezproxy-f.deakin.edu.au/10.1071/AH16287 Li JY, Yong TY, Hakendorf P, Ben-Tovim DI, Thompson CH. Identifying risk factors and patterns for unplanned readmission to a general medical service. Australian Health Review. 2015 Feb 24;39(1):56-62.
--	--

VERSION 1 – AUTHOR RESPONSE

Reviewer 1	
Studies usually distinguish between early readmissions and late readmissions after discharge. It is generally assumed that early readmissions, like 30-day post-discharge, is a better signal of quality of care received for the index hospitalization.	Thank you for this point, the authors agree with this point. The OPTIMAL project was planned to provide a productive model and effective intervention for early readmission. However, to understand the characteristics of patients at risk of frequent admissions, we looked at all patients who were admitted in the last 12 months as looking at only frequent admitters within 30 days would have provided too small a sample size to understand the characteristics. In our sample, only 32/347 of patients who responded to the survey were readmitted within 30 days.

The study does not make the distinction between planned and unplanned readmission. Is it possible to distinguish this and include cases of only unplanned readmission rather than combine both planned and unplanned readmissions?	Thank you for highlighting this point. The authors agree that this would be valuable information, however for this phase of the project this data was not collected as part of the survey used for data collection, hence we are unable to distinguish the cases for this manuscript.
Does the study only consider individuals who were admitted to the same hospital? Do we know anything about individuals who might be readmitted to other hospitals within the same area and if this affects the results of the analysis?	The data collection for this study specifically requested patients to comment on their admissions to CUH. However, patients were not asked about admissions to other hospitals within the area, therefore patients may have reported a greater number of admissions but these were not reflected in the data analysis completed as part of our study. This has been added as a limitation, please see page 3, line 15.
Page 7, line 19 delete the researchers	Thank you for highlighting this. We have now amended the manuscript.
Page 8, can length of stay during index hospitalization be included as an independent variable in the regression analysis? Length of stay is generally used to predict readmissions.	The authors appreciate this comment and agree that length of stay is a predictor of readmission. Unfortunately, the length of stay variable was not recorded during the initial phase of this study. Postal surveys were sent to patients based on their most recent admission. These surveys collected patient reported data on previous admissions within a 12 month period, however length of stay data was not exported from the hospital reporting system for each reported patient admission and we are therefore unable to comment or include in the analysis.
Similarly, can regression analysis be run for elder patients (for e.g. 55 and above) only? Given the mean age of 69.2, individuals with low age might be outliers in the data.	Thank you for making this suggestion. The authors have run an analysis for those patients ≥ 55 years of age. The OLS regression run for this sample provided an f-statistic of 0.007, which we deem to be significant (<0.05). Age (0.003), being in receipt of care (0.004) and admission due to a fall remained significant (0.001). In addition, admission due to infection also became significant (0.002). A statement in the results has been added, please see page 10, line 18-20.
Page 8, Line 59, /333 is missing for the row Prefer not to say.	Thank you for bringing this to our attention. The tables, including this missing row, have now been amended. Please see Table 1.
Page 10. On the regression results are the coefficients meaningful?	The regression coefficients represent the contribution of the independent variable towards predicting the dependent variable. The coefficients are the values that multiply the independent variables and the sign

	represents the direction of the relationship. We consider the regression coefficients of the variables to be meaningful when the p-value is less than or equal to 0.01. With this as the guideline the following variables A_Q1_Adm_Reason_Code_1.0 (falls) (p-value = 0.01), D_Q4_Gender_1.0 (male) (p-value = 0.007), D_Q8_IsComplex_0.0 (Not complex) (p-value = 0.003), D_Q19_Some_Care_1.0 (In receipt of social care)(p-value = 0.005) and ptn_age (p-value = 0.008) are statistically significant.
Reviewer 2	
TITLE and ABSTRACT There is some confusion for the reader in the term ‘frequently readmitting’ in the title and the inclusion criteria for the study cohort. Please confirm whether it is ≥ 2 admissions or ≥ 2 readmissions in 12 months. The aims suggest that participants had at least 3 admissions, but the inclusion criteria in participants section clearly state ≥ 2 admissions. The results also show the range for number of admissions started at 2. This is an important point for clarification because 2 admissions in 12 months would not normally be regarded as frequent.	Dear reviewer, the survey was sent to any CUH patient who had ≥ 2 admissions to CUH in the previous 12 months. However, the linear regression was conducted on those patients who had >2 admissions in the previous 12 months. Therefore, we considered patients with >2 admissions (meaning 3 or more) to be frequent admitters in line with the literature. We have checked the text to clarify this distinction throughout the manuscript and we appreciate you bringing this to our attention. The abstract and title have now been amended accordingly.
P2 lines 45- 52 – See comments below about regression analysis in the Results section.	Thank you for your comments, these have been addressed in the responses below.
INTRODUCTION P5, line 29 - This study is described as a phase of the OPTIMAL Project, funded by Innovate UK. A reference should be provided, either to the protocol for the OPTIMAL project, or a weblink to the funding body, where project is referenced.	Thank you for bringing this to our attention. The reference for the OPTIMAL project has now been added. Please see reference 15 with the associated weblink.
P5 Line 45 – frequent admitters (≥ 2 in 12 months), not readmissions?	The aim has been amended to remove any confusion throughout the manuscript, with frequent admissions being clarified as admission, not readmissions.
P6, lines 50-59 – Surveys included questions about the patient’s discharge experience and post-discharge experience. It is not clear which	The authors appreciate that this was not clearly outlined in the manuscript. Thank you for bringing this to our attention. The questions were related to the patients’ most recent admission. This has now been clarified in the methods, please see page 7, line 3.

admission / readmission the survey participant were asked about?	
P7, lines19 and 21, the term 'the researchers' is repeated.	Thank you for highlighting this. We have now amended the manuscript accordingly.
P7 line 42 – suggest replacing the word 'complaints' with comorbidities, as that is what has been described in the preceding section.	The authors appreciate this suggestion and have made the amendment. Please see page 7, line 20.
P7 line56 to P8 line 19 – If the regression was conducted as described this is not linear regression, but logistic regression. Logistic regression is used to model a binary dependent variable (in this case 'frequent-admitter' –Yes or No). The definition of 'frequent admitter' is described as a patient who had 3 or more admissions in 12 months, compared to a non-frequent admitter who had 2 admissions in 12 months. This comparison is possibly too small a difference for logistic regression, especially as the final number of patients included in the regression analysis was 169, for 7 independent variables. Further comments about the regression analysis are provided in results section. Perhaps this data was not recorded, but were the readmissions for the same diagnosis / condition as the index admission? What was the median time between admissions? What proportion of readmissions were within 30 days?	The authors agree that this is a binary regression however we did not model it as a logistic model but rather as a linear probability model (ordinary least squares). Since the aim of this regression was to study the relevance of the independent variables and the dependent variable rather than build a predictor, we felt that this type of regression would suffice (r-squared score = 0.18). There is some evidence that OLS is a reasonable model even when output is essentially binary (https://academic.oup.com/aep/article/41/3/329/5530153) Unfortunately, that data was not recorded and therefore the authors are unable to comment on the suggested points of interest. However, only 32 (10.8%) of responding patients were admitted within 30 days.
RESULTS The format of the tables could be improved. Generally horizontal lines are not required throughout, just at the top and bottom of the header and table. The columns are for the parameter / variable and then number and percentage (n, %) or mean and standard deviation (mean, SD) or median and interquartile range (mdn, IQR). It was noted that the denominator changed frequently for	The tables have now been amended as suggested, thank you for proposing this amendment to the format which the authors agree was necessary for improvement.

different items, and when this occurs, the number relevant for that item can be written after the variable name.	
In Table 1, Ethnicity, the number and percentage for 'Prefer not to say' was not in the same format as items above. Throughout the results, the denominator used to calculate percentages changed frequently. I assume this was due to missing data when respondents did not answer some questions. This means that some statements misrepresent the true percentage of all respondents. For example, the total number of surveys returned was 347, the number reporting age is 334, gender is 337, ethnicity 333 etc..... In most instances, the authors have made this clear. However, in the section on Discharge, the statements are confusing. It would be much clearer if the same denominator was used throughout this section – either 347 as total number of respondents, or 334 as the number who completed the questions relating to discharge. In particular, the statement on line 46 – 'nearly three quarters of patients (72.4%, n=231) were not provided with contacts for out-of-hours support' is misleading as $231/347 = 66.6\%$. This misrepresentation of percentages is also evident in Table 2; the number of respondents who were told who to contact if their health deteriorated was 84, that is 24.2% of all respondents, but is reported as 70.6% (84/119). It is not clear why 119 was used as the denominator for some items in Table 2, but this over-represents the proportion of respondents for those items. The item below this in the table is written in the negative – 'Not told who to contact for our-of-hours help', which is inconsistent with the item above. Please review and revise this table for consistency.	Table 1 was left as valid given that some patients prefer not to report demographic details. However, the authors have taken your suggestion into account and have therefore amended the tables to use True percentages unless otherwise stated with an alternative N number. Negative wording has been corrected in Table 2 and any amendments to these numbers and percentages have been reflected in the manuscript to ensure consistency. Thank you for bringing this to our attention.
Similarly, in the section on Medications, the number of patients (N) is recorded as 236. But this does	Dear reviewer, following your comprehensive review and suggestions regarding results and demonstration of data, we recompleted the analysis for the entire paper,

not match the text, where the first sentence has N=341, and even removing the 13 patients who reported taking no medications, does not result in N=236? Perhaps the denominator for Table 3 could be the total number of medications reported.	including the linear regression and items such as the medications summary. Following completion of the analysis, we have amended the results in the tables to reflect the updated data, which has also been reflected in the manuscript's text. Please see Table 3 and 4.
The median time to readmission from index admission and the discharge destination were not reported, but are of interest to the reader.	Dear reviewer, unfortunately this data was not collected for this phase of the study and therefore the authors are unable to comment.
P10 lines 14-21 Regression analysis- there is insufficient detail provided to assess the validity of the results from the regression analysis. No univariate analysis has been provided for the items included in the regression. If the regression was conducted as described in the methods, the items which were found to be significant were for increased risk of hospital admission (compared to those who had 2 admissions in 12 months). As such, the variables of admission related to a fall, being in receipt of care at home are logical. However, the item for not having a complex condition or reporting less than 2 conditions being associated with increased risk of hospital admission is not logical. For the regression analysis to be included, further details need to be provided.	Thank you for this point. Based on your advice we ran an additional univariate analysis and found that age, CPS and no. meds were strongly correlated with the frequency of admission with a positive coefficient (F-Score close to 0). As far as we know, there is no guarantee that significance (or non-significance for that matter) of a variable translates from univariate to multiple regression. It has been used as a rule of thumb in the past to check what variables to include in multiple regression but even this is disputed: https://www.annemergmed.com/content/methstats-multivariable_variable_selection In multiple regression the 'question' being asked changes from simple correlation to finding correlation while controlling for the other independent variables in the regression. It could just be that there is not sufficient data to bring out the relation for those variables in the multiple regression scenario. Nevertheless, when we ran the regression analysis for those >3, CPS became significant (p=0.002), which has been added to the results, please see page 10, line 21. Interestingly, for those patients in the regression analysis for admissions >2, CPS scored 0.02, however we were using <0.01 as a means to determine significance.
P13, line8-12 – Post-discharge experience - It was surprising that in a series of statements of results from the survey, one p-value is given. This suggests that Chi-square analysis was done comparing groups of respondents who were / were not	Thank you for bringing this to our attention. The authors felt that it was important to convey that despite medication counselling being provided, this resulted in increased confidence in managing healthcare issues, but did not result in changes to confidence with respect to social-care issues.

confident in managing their own health. It is difficult to assess the validity of this statistical result with the limited information provided.	
P13, line 45; suggest this statement about referral to HOT clinic be prefaced by stating the number of patients this would have been an appropriate, relevant referral, as described in the discussion.	Thank you for bringing this to our attention. The sentence has been amended to reflect the points made in the discussion. Please see
DISCUSSION While the discussion is clearly written, the description of the study findings in relation to current literature was limited.	Dear reviewer, Thank you, we hope that the authors have clearly addressed your points added below.
P14, lines 14-16: To the authors knowledge this may be the first study to capture the complete patient journey across all ages. However, there is an Australian study published in 2019 by Berry et al., that looked at this issue for older patients. See full reference below.	The authors appreciate you sharing this paper with us. As the study was conducted in 2017 and written up shortly after, we were unfamiliar with the work. We have provided additional discussion relating to this paper and the similarity of findings, particularly with respect to comorbidity and polypharmacy. The authors do however comment that they felt this manuscript has approached a broad patient sample across the entire patient journey and is hence novel in this regard. Please see page 16, line 12-17.
P15 , lines 17-32 – ‘in this study medications and conditions (comorbidities) ...did not act as predictors of readmission.’ This would seem to be contrary to other published studies, such as that by Considine et al., 2019 that found the presence of comorbidities was a significant feature for adult medical patients and when controlled for confounders, the number of comorbidities increase the risk of unplanned readmissions within 28 days. Further a study by Li et al, 2015 on the factors associated with unplanned readmission found a Charlson index ≥ 3 was independently associated with increased risk of 28 day admission. (references below)	Thank you for bringing this to our attention. The authors note the significant association between the Charlson index and readmission (Li et al). We also found the contradictory findings in the paper by Considine et al particularly interesting in regard to comorbidity not being a significant predictor in ≤ 1 day readmissions. As discussed above, the authors recompleted the analysis for >3 admissions in the patient sample and found CPS to be significant for this cohort. The authors note the relationship between pre-index hospital utilisation and the significant factors discussed in the literature, we hope that this better represents our findings within the wider context of the current research area. Please see page 16, line 18-25, page 17 line 1-2.
REFERENCES A very consistent style of referencing has been used, however the in-text citations do not always conform with	We appreciate this being highlighted. All references have been checked in text to conform with the guidelines, thank you for your attention to detail.

the guidelines for authors. For example p14 lines 41 and 45, the in-text reference [22] is not immediately following the punctuation but has a space, and then has no space before the start of the following sentence. Berry D, Street M, Considine J. (in press) Service use by older very frequent emergency department users: A retrospective cohort study. Australasian Emergency Care. Accepted 18th May 2019. Considine J, Fox K, Plunkett D, O'Reilly M, Darzins P. (2019). Factors associated with unplanned readmissions in a major Australian health service. Australian Health Review, 43(1), 1-9. doi:http://dx.doi.org.ezproxy-f.deakin.edu.au/10.1071/AH16287 Li JY, Yong TY, Hakendorf P, Ben-Tovim DI, Thompson CH. Identifying risk factors and patterns for unplanned readmission to a general medical service. Australian Health Review. 2015 Feb 24;39(1):56-62.	
---	--

VERSION 2 – REVIEW

REVIEWER	Sushant Joshi University of Southern California United States of America
REVIEW RETURNED	17-Apr-2020

GENERAL COMMENTS	Thank you for the opportunity to review this manuscript again. The authors have addressed most of the concerns I had with the paper. Below are some comments I would like the authors to address. 1. On regression analysis, in sub-sample analysis of age\geq55 and when patients with admissions >3 the number of observations is not mentioned. Also, the independent variables in the regression analysis were admission reason, ethnicity, condition complexity indicator (which was set if a patient described their existing situation as complex/complicated or reported ≥ 2 conditions), a care indicator (identified by patients who were in receipt of some home care), CPS, patient age, number of medications but results are given only for the variables that were significant statistically. Table 1 provides information on gender, but this was not included in the regression analysis. Was there a reason to omit this variable? For Ethnicity, there are categories with a small number of observations (less than 10). Were all the categories included in the analysis or were these recoded, so the total categories were
--

	reduced. The details on regression analysis are limited and it might be helpful to look at the full regression results. 2. The number of respondents sent questionnaires do not match in the Methods and the Results section. The Methods section notes that this number is 2732 whereas the Results section mentions this as 2722. 3. Table 1: you could combine certain ethnicity and have fewer categories since some of the categories have less than five respondents. As a suggestion, you could have a White, Black, Asian, and Other (combine Chinese, Mixed, Other, and Prefer not to say). 4. The references need to be matched i.e. in body and end of text reference numbers do not match. E.g. in body text we have “The number of medications most associated with adverse drug reactions (ADR) resulting in hospital admission was also recorded for each patient.[17]” but end text reference is “17 Chew LD et al. Validation of screening questions for limited health literacy in a large VA outpatient population. J Gen Intern Med. 2008;23(5):561–6.”
--	---

REVIEWER	Dr Maryann Street Deakin University and Eastern Health, Australia
REVIEW RETURNED	06-Apr-2020

GENERAL COMMENTS	Thank you for revising your manuscript to address the concerns that I raised in the original review. You have undertaken considerable further analysis and clarified many of the results in the text and tables. You have also added to the discussion with updated literature. With regard to the regression analysis, you have suggested that using a linear probability model (ordinary least squares) for the regression with a binary dependent variable will suffice. This may be the case, but not within my experience and therefore I leave that for the editors to determine whether referral to a statistician is appropriate.
--

VERSION 2 – AUTHOR RESPONSE

Reviewer Comment	Response
Editor	
The reviewer(s) have recommended revisions to your manuscript. Therefore, I invite you to	Thank you again for taking the time to compile the comments from the reviewers for our resubmitted manuscript. We appreciate the efforts to support our work and hope that this revision meets the requirements for publication in BMJ Open.

respond to the reviewer(s)' comments and revise your manuscript. Please note that we normally allow a maximum of two manuscript revisions. As such, we urge you to make all the necessary revisions at this stage in an effort to convince the reviewers that your work is suitable for publication in BMJ Open.	
Reviewer 2 (Dr Street)	
Thank you for revising your manuscript to address the concerns that I raised in the original review. You have undertaken considerable further analysis and clarified many of the results in the text and tables. You have also added to the discussion	Dear Dr Street, the authors thank you for your comprehensive assessment of the previous manuscript. Following your comments, the authors made significant efforts to improve our presentation of the results based on your feedback. The authors are happy to hear that you recognise the efforts made in this regard. Thank you for the suggestion for the referral to a statistician if appropriate. We sought advice from the statistician on our team regarding this matter who has provided some references below. The statistician for the team will also be on hand to answer queries in the event that a referral is made, we appreciate the on-going support of this manuscript. - http://folk.uio.no/stvoh1/Q%26Q%20Linear%20vs%20logistic%20regression.pdf - https://www.researchgate.net/publication/51992844_Mostly_Harmless_Econometrics_An_Empiricist's_Companion

with updated literature. With regard to the regression analysis, you have suggested that using a linear probability model (ordinary least squares) for the regression with a binary dependent variable will suffice. This may be the case, but not within my experience and therefore I leave that for the editors to determine whether referral to a statistician is appropriate.	
Reviewer 1 (Sushant Joshi)	
On regression analysis, in sub-sample analysis of age\geq55 and when patients with admissions >3 the number of observations is not mentioned. Also, the independent variables in the regression	Dear reviewer, thank you for bringing this to our attention, the number of observations have now been added. Please see page 8, line 14. There was no reason to omit the gender variable, so thank you for bringing this to our attention. We reran the analysis for >2 and >3 admissions looking at gender and found that male gender was a significant contributor to admissions both >2 ($p=0.007$) and >3 ($p=0.014$) in the previous 12 months. We have added this as a variable in the method and included the p-value in the results section. Please see page 8, line 10 (method), page 10, line 17 (results). This change is also reflected in the discussion, please see page 16, line 1 and page 16, line 5-7. We have added a line in the paper to highlight that all other variables were not significant (Please see page 10, line 22). With regards to ethnicity, although we

analysis were admission reason, ethnicity, condition complexity indicator (which was set if a patient described their existing situation as complex/complicated or reported ≥ 2 conditions), a care indicator (identified by patients who were in receipt of some home care), CPS, patient age, number of medications but results are given only for the variables that were significant statistically. Table 1 provides information on gender, but this was not included in the regression analysis. Was there a reason to omit this variable? For Ethnicity, there are categories with a small number of observations (less than 10). Were all the categories included in the analysis or	ran the analysis for all ethnicities, we did not expect to reach statistical significance due to the small sample size within the minority groups. However, for the white ethnicity the p-value was 0.195. All other ethnicities run individually or grouped within BAME categories were not found to be statistically significant with values nearing 1. The number of medicines was found to be insignificant with a p-value of 0.865. When analysing for CPS for those with >2 admissions, this factor was not found to be significant until we ran the analysis for those with >3 admissions, which is highlighted in the paper as significant ($p=0.02$). Being described as having a complex history (e.g. ≥ 2 conditions) was not found to be significant in both the >2 ($p=0.098$) and >3 ($p=0.138$) admission samples.
--	---

were these recoded, so the total categories were reduced. The details on regression analysis are limited and it might be helpful to look at the full regression results.	
The number of respondents sent questionnaires do not match in the Methods and the Results section. The Methods section notes that this number is 2732 whereas the Results section mentions this as 2722.	Thank you for bringing this to our attention. The correct value is 2722, which was incorrectly reported in the methods. This has now been amended. Please see page 6, Line 10 and Line 13.
Table 1: you could combine certain ethnicity and have fewer categories since some of the categories have less than five respondents. As a suggestion, you could have a White, Black, Asian, and Other	Thank you for this suggestion. The authors appreciate that grouping the ethnicities may better reflect the spread of data within this demographic factor. Table 1 has now been amended, please see page 9.

(combine Chinese, Mixed, Other, and Prefer not to say).	
The references need to be matched i.e. in body and end of text reference numbers do not match. E.g. in body text we have “The number of medications most associated with adverse drug reactions (ADR) resulting in hospital admission was also recorded for each patient.[17]” but end text reference is “17 Chew LD et al. Validation of screening questions for limited health literacy in a large VA outpatient population. J Gen Intern Med. 2008;23(5):56 1–6.”	Thank you for your attention to detail. The authors apologise for the incorrect referencing. Reference 17 was placed incorrectly but is correct in what the authors intended to reflect in this section. This reference has now been moved. Please see page 8, line 1. Some of the latter references were incorrect by 1 digit and have now been amended. Additionally, a surplus reference has now been removed and a new reference added, so there are 40 in total. The authors greatly appreciate you bringing this to our attention. The reference list has been updated, please see page 20, line 1 onward.

VERSION 3 – REVIEW

REVIEWER	Sushant Joshi University of Southern California, United States
REVIEW RETURNED	20-May-2020
GENERAL COMMENTS	Thank you for revising your manuscript. You have addressed the concerns I raised in earlier review adequately.